# Cost-effectiveness of monthly follow-up for the treatment of uncomplicated severe acute malnutrition: An economic evaluation of a randomized controlled trial

Nicolas A. Menzies[1,2]*, Fatou Berthé[3], Matt Hitchings[4], Philip Aruna[5], Muhammed Ali Hamza[6], Siméon Nanama[7], Chizoba Steve-Edemba[8], Ibrahim Shehu[3], Rebecca F. Grais[9], Sheila Isanaka[1,9,10]

1 Department of Global Health and Population, Harvard T. H. Chan School of Public Health, Boston, Massachusetts, United States of America, 2 Center for Health Decision Science, Harvard TH Chan School of Public Health, Boston, Massachusetts, United States of America, 3 Epicentre Niger, Niamey, Niger, 4 Department of Biology and Emerging Pathogens Institute, University of Florida, Gainesville, Florida, United States of America, 5 Médecins sans Frontières–Operational Center Amsterdam, Amsterdam, Netherlands, 6 Sokoto State Nutrition Office, Sokoto, Nigeria, 7 UNICEF West and Central Regional Office, Dakar, Senegal, 8 UNICEF Nigeria, Abuja, Nigeria, 9 Epicentre, Paris, France, 10 Department of Nutrition, Harvard T. H. Chan School of Public Health, Boston, Massachusetts, United States of America

* nmenzies@hsph.harvard.edu

**Data Availability Statement:** The deidentified dataset supporting this research can be made

## Abstract

Severe acute malnutrition (SAM) is a major source of mortality for children in low resource settings. Alternative treatment models that improve acceptability and reduce caregiver burden are needed to improve treatment access. We assessed costs and cost-effectiveness of monthly vs. weekly follow-up (standard-of-care) for treating uncomplicated SAM in children 6–59 months of age. To do so, we conducted a cost-effectiveness analysis of a cluster-randomized trial of treatment for newly-diagnosed uncomplicated SAM in northwestern Nigeria (clinicaltrials.gov ID NCT03140904). We collected empirical costing data from enrollment up to 3 months post-discharge. We quantified health outcomes as the fraction of children recovered at discharge (primary cost-effectiveness outcome), the fraction recovered 3 months post-discharge, and total DALYs due to acute malnutrition. We estimated cost-effectiveness from both provider and societal perspectives. Costs are reported in 2019 US dollars. Provider costs per child were $67.07 (95% confidence interval: $64.79, $69.29) under standard-of-care, and $78.74 ($77.06, $80.66) under monthly follow-up. Patient costs per child were $21.04 ($18.18, $23.51) under standard-of-care, and $14.16 ($12.79, $15.25) under monthly follow-up. Monthly follow-up performed worse than standard-of-care for each health outcome assessed and was dominated (produced worse health outcomes at higher cost) by the standard-of-care in cost-effectiveness analyses. This result was robust to statistical uncertainty and to alternative costing assumptions. These findings provide evidence against monthly follow-up for treatment of uncomplicated SAM in situations where weekly follow-up of patients is feasible. While monthly follow-up may reduce burdens on caregivers and providers, other approaches are needed to do so while maintaining the effectiveness of care.

available following a submitted request as per Epicentre and General Data Protection Regulation (EU) 2016/679 data sharing policy. Additional information is available at https://epicentre.msf.org/index.php/en/ongoing-projects/study-data-access-request.

**Funding:** This research was funded by the Children's Investment Fund Foundation (to SI). The funders had no role in study design, data collection and analysis, decision to publish, or preparation of the manuscript.

**Competing interests:** The authors have declared that no competing interests exist.

## Introduction

Undernutrition is a major cause of pediatric illness and mortality [1], and in 2019 was estimated to cause 7% (6%, 9%) of all disability-adjusted life years among children aged 1–4 years old [2]. Severe acute malnutrition (SAM)—characterized by weight-for-height <3 standard deviations below WHO growth standards, mid-upper arm circumference (MUAC) <115mm, and/or nutritional oedema—represents a significant fraction of malnutrition-related deaths and a major mortality risk for affected children [3]. Children surviving SAM experience higher rates of morbidity, mortality, and other adverse outcomes later in life [4, 5]. In 2020, an estimated 13.6 (10.6, 16.7) million children under 5 developed SAM [6]. The current approach for treating SAM in low resource settings combines outpatient treatment with ready-to-use therapeutic foods (RUTF) for uncomplicated cases and inpatient treatment for stabilization of complicated cases. Early published studies of community-based SAM management showed this model to be effective [7–10], and recent systematic reviews [11, 12] have found this approach to be equally effective and substantially less costly compared to earlier approaches that required routine hospitalization for all SAM cases. However, this community-based approach can be difficult to implement in some settings, with the frequent clinic visits required for outpatient care—generally conducted weekly or biweekly until program discharge—increasing the burden on both service providers and caregivers. The challenges related to frequent clinic visits are particularly acute in high-burden, low-resource settings, where provider capacity is often limited and patients may face substantial difficulties to attend frequent clinic visits [13].

It is possible that effective outpatient treatment of uncomplicated SAM can be achieved with fewer scheduled clinic visits, thus reducing costs borne by caregivers and healthcare providers. Under a monthly follow-up schedule, the gap between scheduled clinic visits would be longer and caregivers would be given larger therapeutic food rations at each visit, as well as additional education and support when initiating treatment. A pilot study of this approach conducted in Niger found that adequate treatment response and use of therapeutic foods was maintained when following a monthly schedule of follow-up [14]. In the current study, we conducted a cost-effectiveness analysis as part of a cluster randomized trial of community-based SAM treatment [15] to estimate the costs and cost-effectiveness of a strategy of monthly outpatient follow-up for children diagnosed with uncomplicated SAM, as compared to weekly outpatient follow-up (the current standard-of-care).

## Materials and methods

### Study setting and population

This study was conducted in Sokoto state, northwestern Nigeria, a rural area with endemic acute malnutrition. In 2018, SAM prevalence in Sokoto was estimated to be 7.9% among children under 5 years old, the highest rate of malnutrition in the country [16]. The study population included children 6–59 months of age who were newly admitted to treatment for uncomplicated SAM (mid-upper arm circumference (MUAC) < 115 mm and/or grade 1–2 edema and absence of current illness requiring inpatient care) at one of 10 outpatient therapeutic feeding centers (OTP) in Binji and Wamako local government areas, between January 23, 2018 and November 30, 2019. Management of SAM was provided free of charge in all 10 study clinics by the Sokoto State Ministry of Health, with support from UNICEF. There was no MAM treatment program operational in the study site at the time of enrollment.

### Intervention and clinical trial

Participants received one of two intervention strategies: a "monthly" follow-up strategy, in which participants attended clinic visits at weeks 4, 8, 10 and 12 from enrollment until

discharge with caregiver support at admission for continued home-based surveillance [15, 17], or a weekly follow-up strategy (the current standard-of-care), in which participants attended standard weekly clinic visits from enrollment to discharge. Scheduled OTP visits included medical and anthropometric surveillance and provision of therapeutic feeding rations. All participants received a home visit by a community health worker when a scheduled clinic visit was missed to ascertain the reason for absence and encourage the caregiver to return to treatment. As per study protocol, data to assess sustained recovery were collected at a 3-month post-discharge visit (the costs of this post-discharge visit were not included in the cost analysis).

Randomization was conducted at the level of the OTP with a crossover design. The 10 OTPs were stratified into 5 groups by size and assigned randomly 1:1 to the monthly follow-up or standard-of-care arms. Prior to crossover, all children attending a given OTP received the same schedule of follow-up. Following crossover (December 17, 2018), each site switched from monthly to weekly follow-up, or vice-versa. Children enrolled before the cross-over continued to follow the strategy they were enrolled on. The target sample size in each study arm (1750, with 175 per OTP) was calculated to achieve >90% power to detect a noninferiority margin difference between the group proportions of −0.10. A detailed description of trial design and sample size calculations are provided in Hitchings et al. [15]. This study was registered at clinicaltrials.gov (ID NCT03140904).

## Cost-effectiveness analysis

The primary cost-effectiveness endpoint was the incremental cost per child achieving nutritional recovery under the monthly follow-up strategy, as compared to a weekly follow-up (standard-of-care) strategy, from both provider and societal perspectives. We also estimated the cost per child treated and the cost per child recovered under each study arm, as well as detailed costing results for each intervention activity. In secondary analyses, we report the cost per child recovered based on outcomes at 3-months post-discharge and the cost per DALY averted based on additional assumptions about survival after trial completion. All costs and health outcomes were assumed to occur within a year of intervention enrollment, and no discounting was applied. We followed published standards for the conduct of the cost-effectiveness analysis [18, 19], and reported results using the Consolidated Health Economic Evaluation Reporting Standards (CHEERS) guidelines [20].

## Cost data

Provider costs were calculated as the resources required to deliver the intervention, excluding research costs. These include the costs of enrollment consultations, education sessions (only for the monthly follow-up arm), follow-up consultations (scheduled and spontaneous), units of therapeutic food provided, dispensed drugs, home visits to trace children after missed scheduled consultations, and inpatient bed-days. For each study participant, we recorded the number of items and services received, multiplied these quantities by the unit cost for each category, then summed across all categories to estimate total provider costs. Costs for enrollment consultations, education sessions and follow-up consultations were based on the cost-per-minute for clinical care to allow for differences in the duration of these services between study arms. This cost-per-minute ($0.42) was calculated by dividing a published outpatient visit unit cost (comprising personnel, infrastructure, maintenance and managerial overheads, and other operational costs) [21] by the average visit duration. For each visit, the visit duration in minutes was calculated as the difference between the start (caregiver/child enter consultation area) and end (caregiver/child exit consultation area, paperwork is complete, area is prepared for

next patient) times of the clinical encounter. The average visit duration was calculated as the sum of these estimates divided by the number of visits. In scenario analyses we examined alternative unit cost estimates. Unit cost estimates are shown in Table 1.

Patient costs were defined as disease-related costs borne by the families of study participants during follow-up. These were calculated as the out-of-pocket spending reported by caregivers as a consequence of seeking care, including care obtained from providers not included in the study, as well as the opportunity cost of caregiver time spent attending clinic visits and during a child's hospitalization. Patient costs associated with outpatient care were collected through a costing sub-study conducted among caregivers. For each caregiver included in this costing sub-study, interviews were conducted at all scheduled clinic visits, as well as at 3-month post-discharge follow-up visit to collect costs incurred after treatment. Patient costs for inpatient care were collected through a sub-study conducted among caregivers of hospitalized patients, with interviews conducted daily during the period of hospitalization. Out-of-pocket costs included medical costs (medicines, diagnostic tests, other care) and non-medical costs (transportation, food, care for dependents, and other items). In the clinical trial, patients

**Table 1. Inputs for cost-effectiveness analysis.**

| Input | Value[¶] | Source |
|---|---|---|
| Unit cost for enrollment consultations | | Stenberg 2018, study data |
| *Standard-of-care arm* | $0.97 | |
| *Monthly follow-up arm* | $0.88 | |
| Unit cost for follow-up consultations | | Stenberg 2018, study data |
| *Standard-of-care arm* | $0.31 | |
| *Monthly follow-up arm* | $0.33 | |
| Unit cost for education sessions | | Stenberg 2018, study data |
| *Standard-of-care arm*[†] | — | |
| *Monthly follow-up arm* | $5.38 | |
| Unit cost for home visits | $0.93 | Study data |
| Unit cost for hospitalization (per bed-day) | $8.48 | Stenberg 2018 |
| Unit cost for therapeutic food (per sachet) | $0.33 | Study data |
| Unit cost for dispensed drugs (per course) | | MSF Logistics Catalogue 2019, UNICEF Supply Division 2019 |
| *Albendazole* | $0.03 | |
| *Measles vaccination* | $0.17 | |
| *Vitamin A* | $0.03 | |
| *Amoxicillin* | $0.14 | |
| Nigerian minimum wage (per hour)[‡] | $0.48 | Government of Nigeria 2019 |
| Exchange rate, Nigerian Naira to US Dollars, 2020 | 358 | Oanda.com |
| Risk of malnutrition-related mortality by outcome recorded at 3 months post-discharge[§] | | Isanaka 2019 |
| *Recovered* | 0.0016 | |
| *Non-recovered (alive)* | 0.0204 | |
| *Non-recovered (died)* | 1.0000 | |
| DALYs (years of life lost) per malnutrition death | 85.21 | Global Burden of Disease Collaborative Network 2019 |

[¶] Unit costs reported in 2019 USD.

[†] Education sessions not conducted for standard-of-care arm.

[‡] Monthly minimum wage divided by 173 working hours per month.

[§] From beginning of intervention until 12 months following post-discharge visit.

were reimbursed for out-of-pocket medical and transportation costs for hospital care. These were included as patient costs for the cost analysis, as they would not typically be reimbursed during routine care in this setting. The opportunity cost of caregiver time was calculated by summing the number of minutes spent by caregivers traveling to or attending clinic and hospital care over the course of treatment, and multiplying by a unit cost per hour, which was based on the Nigerian national minimum wage (Table 1). All costs were inflated to 2019 values using the GDP deflator as a price index [22], and converted into US dollars using market exchange rates.

## Effectiveness outcomes

Nutritional recovery (used to report the cost per child recovered) was defined at or before 12 weeks from admission as being free from medical complications, MUAC $\geq$ 125 mm for 2 consecutive visits, and no edema if admitted with edema at study discharge. Children who defaulted (defined as 3 missed scheduled clinic visits in the weekly follow-up group and 1 missed scheduled clinic visit in the monthly follow-up group), transferred to inpatient care, died, or were lost to follow-up for other reasons before study discharge were treated as not recovered. We also estimated outcomes using an alternative definition of recovery based on outcomes at 3-months post-discharge. This outcome, sustained recovery, required that children be classified as recovered at study discharge (as described above), and additionally to have not died, been hospitalized, or relapsed by the 3-month post-discharge visit. S1 Table provides summary results for major effectiveness endpoints, as reported in Hitchings et al. [15].

In addition to these empirical outcomes we estimated the average number of disability-adjusted life years (DALYs) due to SAM for individuals in each intervention arm, based on methods described in Isanaka et al. [23]. Under this approach we considered observed mortality during the trial (up to the 3-month post-discharge visit) as well as expected mortality after the trial until 12 months post-enrollment, which was calculated by imposing different disease-specific mortality rates depending on whether an individual was classified as recovered by the 3-month post-discharge visit. These were multiplied by life expectancy estimates taken from the Global Burden of Disease Study reference life table [24] (Table 1). All mortality attributable to the SAM episode was assumed to occur in the 12 months following study enrollment.

## Statistical analyses

The incremental cost per child recovered was calculated as the difference in the mean cost per participant between study arms, divided by the difference in the mean probability of recovery between study arms (monthly follow-up compared to standard-of-care). For the provider perspective, we included provider costs incurred between study enrollment and discharge. For the societal perspective, we included these costs as well as patient costs incurred over the same period. For the alternative outcomes with a longer analytic horizon (sustained recovery by 3-month post-discharge and DALYs), we additionally considered patient and provider costs incurred between discharge and the 3-month post-discharge visit. We estimated uncertainty in analytic outcomes using a non-parametric bootstrap at the cluster level [18, 25] with 10,000 iterations, and reported this uncertainty with equal-tailed 95% confidence intervals, two-sided p-values for differences between study arms, scatterplots on the cost-effectiveness plane, and cost-effectiveness acceptability curves [26, 27]. All analyses were conducted with R (version 4.0.3) [28].

## Scenario analyses

We tested the robustness of results to several alternative assumptions: (1) exclusion of home visit costs, as these home visits may not be provided outside of the study setting; (2) a lower

cost per minute for clinic visits ($0.02 per minute, based on clinical staffing costs alone); (3) a higher cost per minute for clinic visits ($1.37 per minute, based on an earlier study of community-based management of acute malnutrition in northern Nigeria [29]); (4) a lower hospital bed-day cost ($1.72 per day, based on an earlier study of community-based management of acute malnutrition in northern Nigeria [29]); (5) drug dispensing assumed to follow the national protocol, whereby each child receives albendazole, amoxicillin, and vitamin A, and additionally assuming that 50% would also get 1 dose of measles vaccine; (6) a lower cost per minute applied to education sessions ($0.02 per minute, based on clinical staffing costs alone); and (7) a lower cost per minute applied to education sessions ($0.02 per minute) and additionally assuming that therapeutic food costs in the intervention arm would be the same as in the control arm. Total cost estimates were recalculated under each of these seven alternative assumptions.

## Ethics statement

The study was reviewed and approved by research ethics committees at the Harvard T.H. Chan School of Public Health (reference number: IRB17-0221) and the Sokoto State Ministry of Health, Nigeria (reference number: SMH/1580/V.IV). An independent data and safety monitoring board monitored study progress. Caregivers of study participants provided written informed consent before admission and were made aware of their ability to withdraw from the study at any time.

## Results

There were 3788 children enrolled in the clinical study with a recorded outcome at program discharge (1802 in the standard-of-care arm, 1976 in the monthly follow-up arm) and 3594 children with an outcome recorded at study discharge at 3 months post-program discharge (1721 in standard-of-care, 1873 in monthly follow-up). Of these children, 1031 were included in the outpatient costing sub-study and 686 in the inpatient costing sub-study. Additional details are provided in Fig 1.

### Effectiveness outcomes for cost-effectiveness analysis

Full results of the effectiveness analysis are described in Hitchings et al. [15]. The standard-of-care performed better for each effectiveness outcome. The proportion of all children recovered by intervention discharge was 58.8% (95% confidence interval: 54.5, 63.1) under standard-of-care and 52.4% (49.0, 56.3) under monthly follow-up, a difference of 6.3 percentage points (2.9, 10.0, p<0.001). The proportion of all children classified as recovered at 3 months post-discharge was 54.6% (51.1, 58.3) under standard-of-care and 49.3% (45.5, 53.1) under monthly follow-up, a difference of 5.3 percentage points (2.2, 9.0, p<0.001). We estimated 6.0 (4.9, 7.1) DALYs per child under standard-of-care and 8.0 (6.7, 9.4) under monthly follow-up, a difference of 2.0 (1.0, 3.1, p<0.001) DALYs per child.

### Service utilization and provider costs

Among children who recovered, the time between enrollment and program discharge was greater in the monthly follow-up group compared to standard-of-care (67.4 days vs. 51.2 days, p<0.001). For the standard-of-care approach, there was an average 8.5 (8.1, 8.9) scheduled consultations (including the enrollment visit) per child, and 4.2 (4.1, 4.3) under monthly follow-up (p-value for difference between arms <0.001). There was little difference in the average number of unscheduled consultations per child (0.32 (0.27, 0.38) for standard-of-care, 0.31

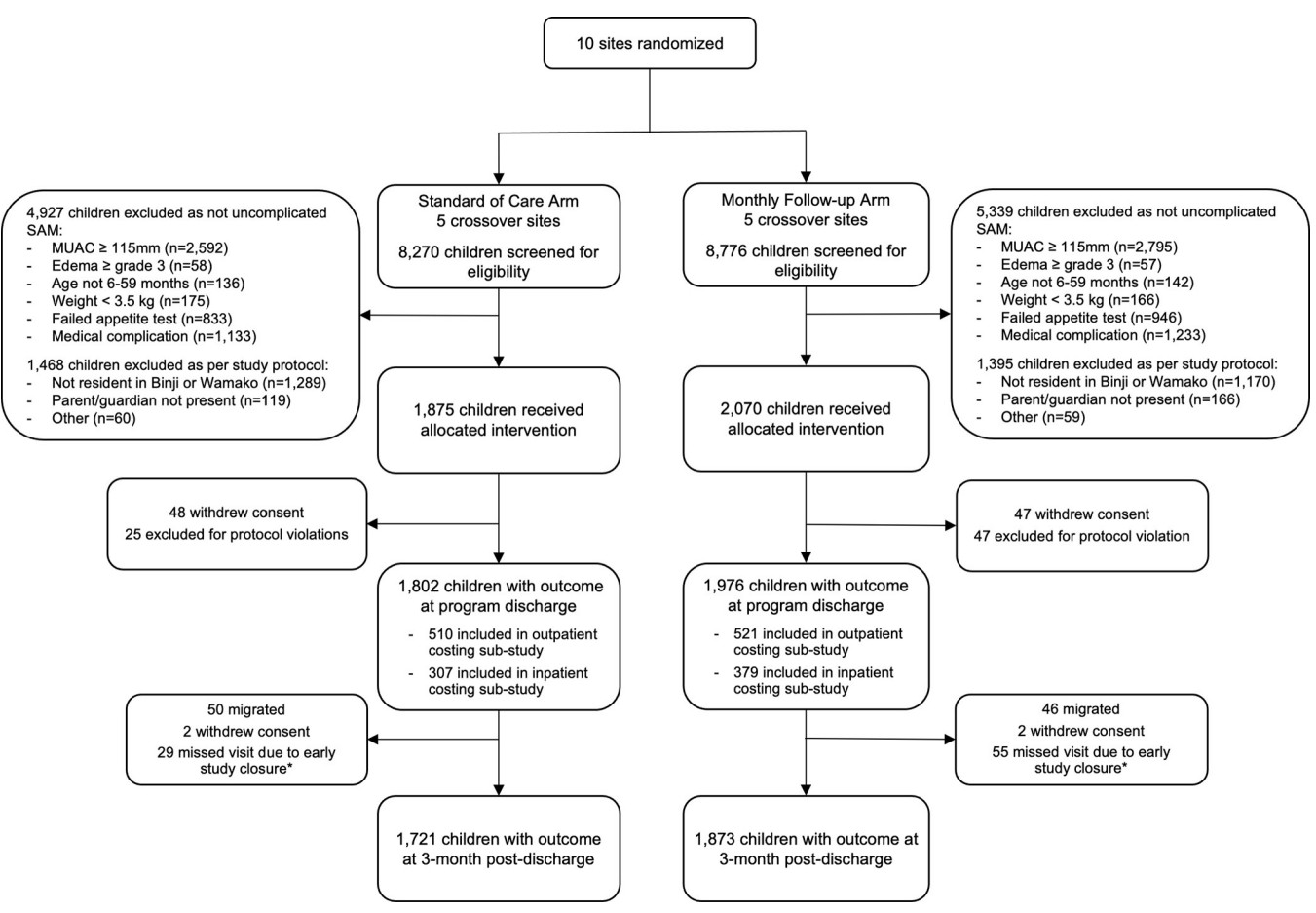

**Fig 1. Flowchart of trial enrollment, randomization and follow up.**

(0.25, 0.40) for monthly follow-up (p = 0.38). Enrollment consultations took an average 9.7 (8.8, 10.6) minutes under standard-of-care, and 10.7 (9.3, 12.8) minutes under monthly follow-up (p = 0.15). Scheduled follow-up consultations took an average 3.4 (3.2, 3.6) minutes under standard-of-care and 3.7 (3.2, 4.3) minutes under monthly follow-up (p = 0.17). Total time taken for consultations (combining number and duration of consultations) was 36.3 (32.9, 37.8) minutes under standard-of-care and 23.8 (20.0, 26.3) minutes under monthly follow-up (p<0.001). Under the monthly follow-up arm caregivers received an education session following the enrollment consultation, on average taking 43.7 minutes (39.4, 48.9). For home visits conducted when a scheduled clinic visit was missed, there were an average 0.52 (0.39, 0.66) visits per child under standard-of-care, and 0.22 (0.17, 0.28) under monthly follow-up (p <0.001). Each home visit took an average of 62.2 (58.5, 66.3) minutes of staff time, including travel to the household. Under the standard-of-care there was an average 147 (141, 153) therapeutic food sachets dispensed during the course of treatment, and an average 184 (180, 187) sachets under monthly follow-up (p<0.001). During treatment there was an average 10.9 (6.6, 15.6) hospitalizations per 100 children under standard-of-care, and 9.5 (6.8, 12.1) under monthly follow-up (p = 0.30). The average duration of hospitalization was 4.0 (2.5, 5.0) days for standard-of-care, and 3.9 (2.8, 5.3) days under monthly follow-up (p = 0.54). Table 2 reports the provider costs associated with these services by cost category and study arm. Total provider costs per child were $67.07 ($65.79, $69.29) for standard-of-care and $78.74 ($77.06,

$80.66) for monthly follow-up (p<0.001). Provider costs per child achieving nutritional recovery were $114.13 ($104.70, $125.04) under standard-of-care and $150.19 ($138.48, $163.05) under monthly follow-up (p<0.001).

## Patient costs

Outpatient costs were estimated from 6374 interviews conducted at scheduled clinic visits and 3-month post-discharge follow-up in a sub-sample of 1054 study caregivers. Across study arms, the average patient out-of-pocket cost per outpatient visit was $0.21 (0.20, 0.22). These costs incurred during study outpatient visits were categorized as transport (91.8%), food (6.4%), out-of-pocket spending for medicines or medical supplies, tests or other procedures (1.4%), dependent care (0.3%), and accommodation (0.1%). Costs incurred seeking care from providers not affiliated with the study (predominantly private pharmacies, mobile pharmacies, and other government-run outpatient clinics) were estimated as $1.17 (0.77, 1.58) per child under standard-of-care, and $1.09 (0.75, 1.54) per child under monthly follow-up. This spending on care outside of study outpatient visits was devoted to purchasing medicines (67.9%), transport (21.1%), food (5.8%), clinical tests (4.9%), and other care (0.2%). Over the course of treatment, caregivers spent an average of 31 (27, 35) hours attending clinic visits under standard-of-care, and 19 (17, 20) hours under monthly follow-up (p<0.001). Across study arms this averaged 3.7 (3.6, 3.8) hours per visit (3.5 (3.4, 3.6) under standard-of-care, and 4.1 (4.0, 4.3) under monthly follow-up, p<0.001). Between discharge and 3-month post-discharge follow-up, caregivers reported incurring additional out-of-pocket costs averaging $0.80 (0.70, 0.89) per child under standard-of-care, and $0.69 (0.59, 0.79) per child under monthly follow-up (p = 0.05). This spending after program discharge was devoted to medicines (68.4%), transport (22.0%), clinical tests (4.5%), food (3.7%), and other items (1.4%). Patient costs from hospitalization were collected through 2322 interviews with a sub-sample of 743 caregivers of hospitalized patients. Across study arms average out-of-pocket costs were $10.06 ($9.26, $10.92) per hospital stay, and $4.20 ($3.95, $4.45) per bed-day. These costs were primarily for medicines, medical supplies, diagnostic tests or procedures (72.1%). Other costs included transport (13.9%), food (12.4%), dependent care (0.8%), and other costs (0.8%). Caregivers in the trial were reimbursed for out-of-pocket medical and transportation costs for hospital care. These were included as patient costs for the present cost analysis. Table 2 reports patient costs by cost category and study arm. Total patient costs per child were $21.04 ($18.18, $23.51) for standard-of-care and $14.16 ($12.79, $15.23) for monthly follow-up (p<0.001). Patient costs per child achieving nutritional recovery were $35.80 ($27.85, $39.61) under standard-of-care and $27.01 ($22.19, $27.22) under monthly follow-up (p<0.001).

Summing provider and patient costs, societal costs per child were $88.11 ($83.61, $91.60) for standard-of-care and $92.91 ($90.58, $95.38) for monthly follow-up (p = 0.03). Societal costs per child achieving nutritional recovery were $149.93 ($135.35, $166.19) under standard-of-care and $177.21 ($164.02, $192.22) under monthly follow-up (p<0.001).

## Incremental cost-effectiveness

Fig 2 compares costs and health benefits of each study arm for the primary cost-effectiveness endpoint (incremental cost per child recovered), under provider and societal perspectives. Compared to standard-of-care, monthly follow-up had higher provider costs (difference $11.67 ($9.35, $13.78, p<0.001)), higher societal costs (difference $4.80 ($0.47, $9.53, p<0.001)), and lower recovery (difference 6.3 percentage points (2.9, 10.0, p<0.001)). With worse health outcomes and higher costs, monthly follow-up was dominated (performed worse

**Table 2. Per child costs by intervention arm and cost category.**

| Cost type | Item description | Average cost per child (2019 USD) | | |
|---|---|---|---|---|
| | | **Standard-of-care** | **Monthly follow-up** | **Difference¶** |
| Provider costs | Outpatient consultations | $ 15.12 (14.14, 16.25) | $ 9.92 (8.72, 11.70) | -$ 5.20 (-7.18, -2.94) |
| | Home visits | $ 0.41 (0.31, 0.52) | $ 0.18 (0.14, 0.23) | -$ 0.23 (-0.33, -0.15) |
| | Education sessions | — | $ 5.38 (4.35, 6.50) | $ 5.38 (4.35, 6.50) |
| | Therapeutic food | $ 48.12 (46.12, 50.33) | $ 60.29 (59.11, 61.43) | $ 12.18 (10.24, 14.04) |
| | Dispensed drugs | $ 0.09 (0.05, 0.11) | $ 0.07 (0.05, 0.09) | -$ 0.02 (-0.06, 0.03) |
| | Hospitalization | $ 3.34 (1.64, 5.29) | $ 2.91 (1.99, 3.65) | -$ 0.43 (-2.64, 1.69) |
| | Post-discharge costs* | $ 0.88 (0.63, 1.17) | $ 0.86 (0.57, 1.18) | -$ 0.02 (-0.48, 0.44) |
| | **Total †** | **$ 67.07 (64.79, 69.29)** | **$ 78.74 (77.06, 80.66)** | **$ 11.67 (9.35, 13.78)** |
| | **Total (incl 3 months post-discharge) ‡\*** | **$ 68.06 (65.86, 70.23)** | **$ 79.80 (78.19, 81.67)** | **$ 11.74 (9.70, 13.52)** |
| Patient costs§ | Outpatient | $ 1.88 (1.10, 2.66) | $ 1.29 (0.94, 1.60) | -$ 0.58 (-1.32, 0.13) |
| | Hospitalizations | $ 1.63 (0.80, 2.57) | $ 1.43 (0.98, 1.80) | -$ 0.20 (-1.25, 0.85) |
| | Other out-of-pocket costs | $ 1.96 (1.59, 2.36) | $ 1.77 (1.41, 2.27) | -$ 0.19 (-0.64, 0.36) |
| | Caregiver time costs | $ 14.84 (12.99, 16.82) | $ 9.02 (8.01, 9.93) | -$ 5.81 (-8.05, -3.75) |
| | Post-discharge costs* | $ 1.23 (1.03, 1.44) | $ 1.11 (0.94, 1.31) | -$ 0.12 (-0.47, 0.20) |
| | **Total** | **$ 21.04 (18.18, 23.51)** | **$ 14.16 (12.79, 15.25)** | **-$ 6.88 (-10.51, -3.33)** |
| | **Total (incl 3 months post-discharge)\*** | **$ 22.12 (19.34, 24.48)** | **$ 15.17 (13.83, 16.23)** | **-$ 6.96 (-10.34, -3.48)** |
| All costs | **Total †** | **$ 88.11 (83.61, 91.60)** | **$ 92.91 (90.58, 95.38)** | **$ 4.80 (0.47, 9.53)** |
| | **Total (incl 3 months post-discharge) ‡\*** | **$ 90.19 (85.78, 93.46)** | **$ 94.97 (92.71, 97.27)** | **$ 4.78 (0.98, 9.06)** |

¶ Difference calculated as value for monthly follow-up minus value for standard-of-care.

† Used for primary cost-effectiveness outcome under provider and societal perspectives.

‡ Used for secondary cost-effectiveness outcomes under provider and societal perspectives.

§ In the clinical trial patients were reimbursed for out-of-pocket medical and transportation costs associated with hospitalization. These were coded as patient costs for the cost analysis, as they would not typically be reimbursed during routine care.

\* Calculation excluded individuals censored before 3-month follow-up.

on both health outcomes and costs) under both provider and societal perspectives, and a cost-effectiveness ratio cannot be calculated.

Similar results were obtained for the secondary outcome of sustained recovery at 3-months post-discharge. For this outcome, monthly follow-up had higher provider costs (difference $11.74 ($9.70, $13.52, p<0.001)), higher societal costs (difference $4.78 ($0.98, $9.06, p<0.001)), and lower sustained recovery (difference 5.3 percentage points (2.2, 9.0, p<0.001)) compared to standard-of-care, and consequently monthly follow-up was dominated under both provider and societal perspectives. Monthly follow-up was also predicted to produce 2.0 (1.0, 3.1, p<0.001) additional DALYs per child compared to standard-of-care and was dominated for this outcome. Fig 3 compares costs and health benefits for the cost per DALY averted. A cost-effectiveness acceptability curve calculated from these results showed that monthly follow-up had <1% probability of being cost-effective for a wide range of willingness-to-pay values, ranging from $10-$10,000 per DALY averted.

## Scenario analyses

Table 3 shows the societal cost per child calculated under alternative analytic assumptions. While Scenario 2 (lower clinical unit cost) and Scenario 3 (higher clinical unit cost) produced substantial decreases and increases in cost estimates (respectively), the incremental differences between study arms were robust to these changes. Scenario 6 (lower education session unit cost) and Scenario 7 (lower education session unit cost and therapeutic food costs assumed to

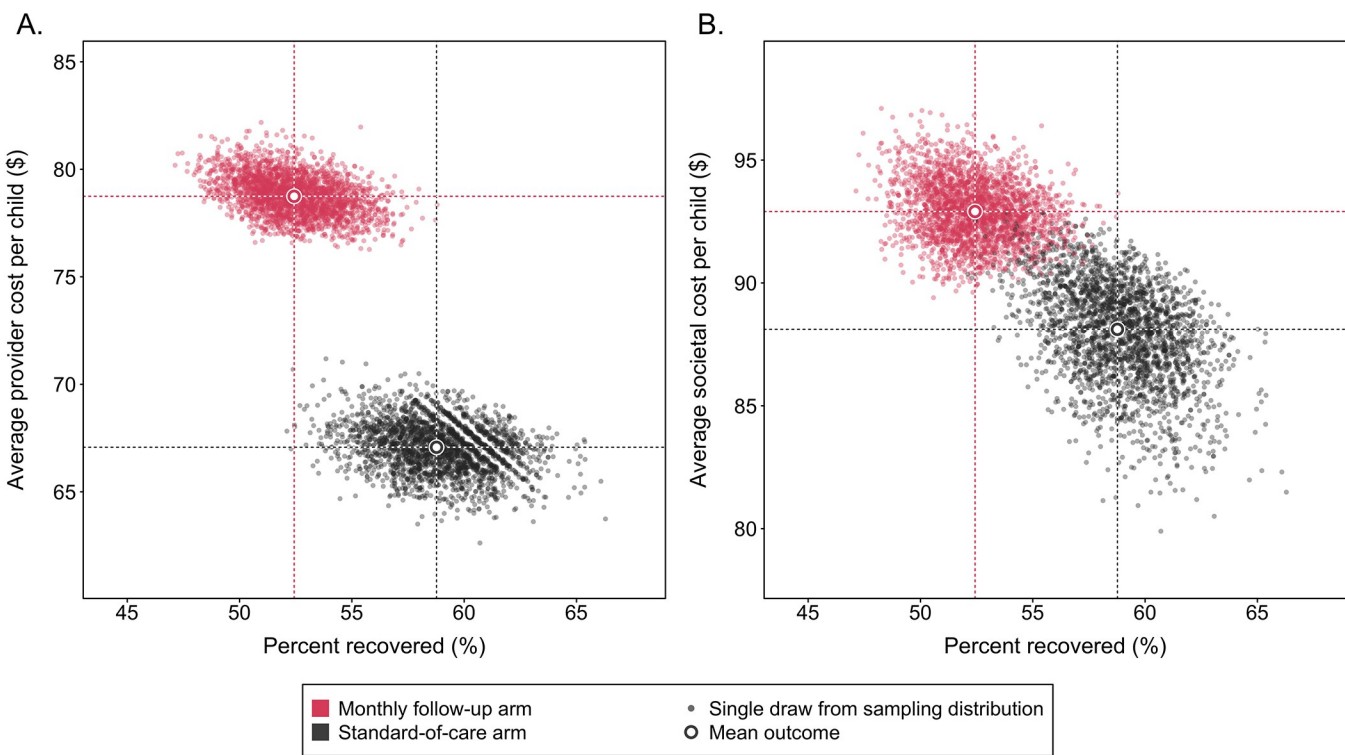

**Fig 2.** Per child costs and probability of achieving nutritional recovery under standard-of-care and monthly follow-up arms, from provider (Panel A) and societal (Panel B) perspectives.

be equal) produced substantial cost reductions for monthly follow-up relative to standard-of-care, such that the per-child cost of monthly follow-up was equivalent to (Scenario 6) and lower than (Scenario 7) the standard-of-care. Other scenarios had a minimal impact on cost estimates compared to the main analysis.

Of all these scenarios, monthly follow-up was not dominated in Scenario 7 (lower education session unit cost and therapeutic food costs assumed to be equal), and therefore a cost-effectiveness ratio could be calculated. Specifically, in this scenario monthly follow-up had lower health outcomes and lower costs compared to standard-of-care, and in this situation the cost-effectiveness ratio describes the cost-effectiveness of the standard-of-care, as compared to monthly follow-up. For this comparison, the incremental cost per child recovered was $198 ($99, $481), the incremental cost per child achieving sustained recovery was $257 ($115, $706) and the incremental cost per DALY averted was $6.76 ($4.12, $13.26).

## Discussion

In this study we examined the costs and cost-effectiveness of a monthly follow-up approach for treatment of uncomplicated SAM in children 6–59 months of age, as compared to a standard-of-care protocol that required weekly clinic visits until program discharge. We found that the monthly follow-up approach was successful in reducing the costs borne by caregivers to receive SAM care for their child, with the number of clinic visits, the total time spent attending care, and total patient costs all substantially lower in the monthly follow-up arm compared to standard-of-care. Monthly follow-up also resulted in lower provider costs for clinical consultations, but it produced higher total provider costs per child, driven primarily by higher costs of therapeutic food, as well as the costs of an educational session that was not provided in

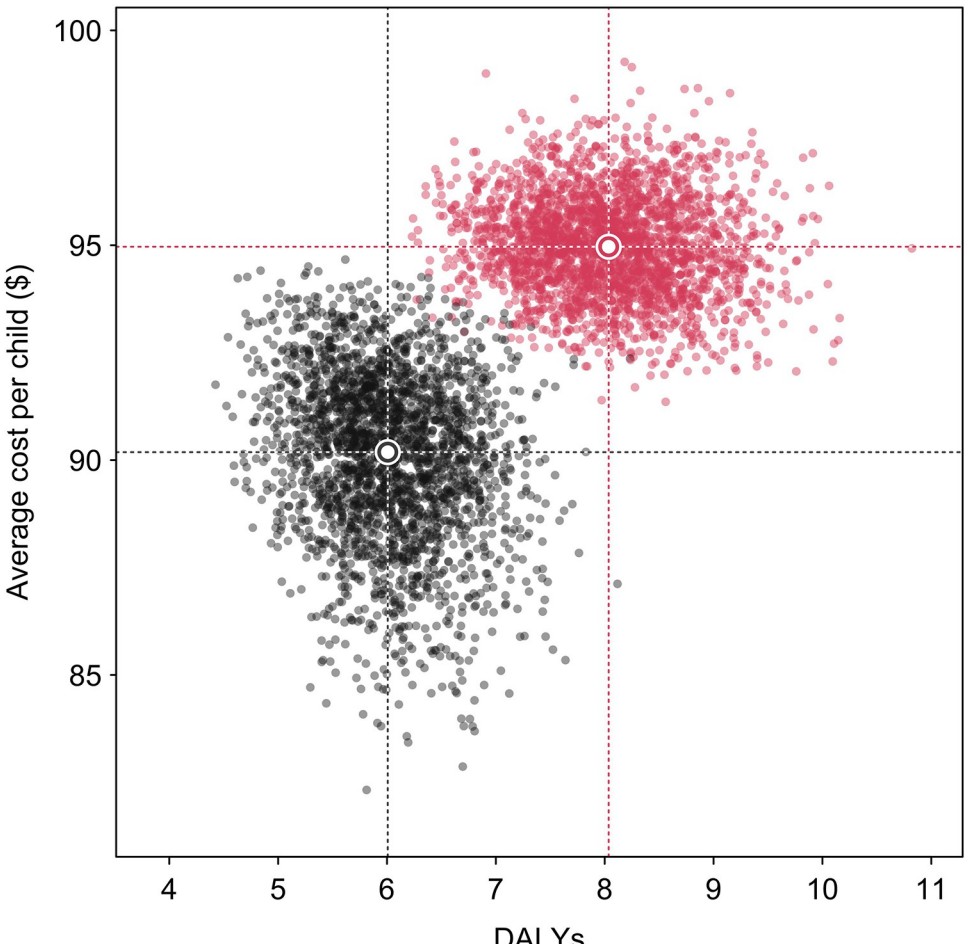

**Fig 3. Per child costs and DALYs under standard-of-care and monthly follow-up arms.**

the standard-of-care. As a result of these differences, the overall cost per child (combining patient and provider costs) was higher for the monthly follow-up arm. Results were similar when we included costs incurred over the 3-months following discharge. The monthly follow-

**Table 3. Estimates of the societal cost per child under alternative analytic assumptions.**

| Assumption scenario | Average cost per child (2019 USD) | | |
|---|---|---|---|
| | Standard-of-care | Monthly follow-up | Difference¶ |
| Main analysis (for comparison) | $ 88.11 (83.61, 91.60) | $ 92.91 (90.58, 95.38) | $ 4.80 (0.47, 9.53) |
| Alternative assumption 1: Exclusion of home visit costs | $ 87.70 (83.19, 91.20) | $ 92.73 (90.41, 95.17) | $ 5.03 (0.72, 9.78) |
| Alternative assumption 2: Lower clinical unit cost | $ 73.56 (69.25, 77.06) | $ 78.19 (75.38, 80.40) | $ 4.63 (-1.16, 10.25) |
| Alternative assumption 3: Higher clinical unit cost | $ 122.52 (116.92, 127.36) | $ 127.72 (122.53, 133.72) | $ 5.20 (-0.50, 11.18) |
| Alternative assumption 4: Lower hospital bed-day cost | $ 85.45 (81.61, 89.03) | $ 90.59 (88.16, 93.06) | $ 5.14 (1.13, 9.11) |
| Alternative assumption 5: Drug dispensing per national protocols | $ 88.38 (83.90, 91.86) | $ 93.20 (90.86, 95.68) | $ 4.81 (0.49, 9.52) |
| Alternative assumption 6: Lower education session unit cost | $ 88.11 (83.61, 91.60) | $ 87.73 (85.82, 89.55) | $ -0.38 (-4.95, 4.36) |
| Alternative assumption 7: Lower education session unit cost, therapeutic food costs equal | $ 88.11 (83.61, 91.60) | $ 75.56 (74.09, 77.33) | $ -12.55 (-16.08, -8.21) |

¶ Difference calculated as value for monthly follow-up minus value for standard-of-care.

up arm was also found to produce worse health outcomes compared to standard-of-care [15], with lower nutritional recovery at program discharge and greater cumulative mortality by 3 months post-discharge. As a consequence, the incremental cost-effectiveness analysis concluded that monthly follow-up was dominated by the standard-of-care, indicating that it produced worse health outcomes at higher cost and should not be adopted under any cost-effectiveness threshold. These results were robust to statistical uncertainty and were qualitatively similar across most of the different alternative assumptions examined in scenario analyses. Even under the most favorable set of alternative assumptions for the costs of monthly follow-up (Scenario 7, with lower unit costs for the education session and therapeutic food costs assumed equal across arms), the incremental cost-effectiveness ratio for standard-of-care compared to monthly follow-up is below $10 per DALY averted, which is well below the threshold at which an intervention would be deemed cost-effective according to conventional criteria [30, 31] and suggests that the standard-of-care should be preferred. Moreover, as adopting monthly follow-up would involve worse health outcomes, some have argued that such changes must meet more stringent criteria than interventions that improve health [32, 33].

An unexpected finding of the cost analysis was that monthly follow-up had higher provider costs than standard-of-care. The major reason for this—greater dispensing of therapeutic food under the monthly follow-up arm—highlights the substantial resources devoted to therapeutic foods, which represent over 50% of total societal costs and over 70% of provider costs across both arms. The reason for greater therapeutic food dispensing with monthly follow-up was the need to provide caregivers with sufficient food during the extended period until the next scheduled visit. Under the standard-of-care, weekly visits allowed more precise control of the amount of therapeutic food dispensed, reducing potential waste. Future modifications to SAM treatment protocols could be designed to make more efficient use of more expensive program inputs, including therapeutic foods. Another notable finding was the substantial costs faced by caregivers to obtain care—representing almost 25% of total societal costs under the standard-of-care arm. Most of these costs attributable to the time spent by caregivers travelling to and waiting for clinical consultations, even though these consultations last less than 5 minutes on average. This result underlines concerns about the burdens placed on caregivers to receive care, one of the motivations for this study. In this study, the cost per child for each study arm was lower than other values reported in other analyses of community SAM treatment [34–36], including an earlier analysis from the same setting [29]. This may be related to the exclusion of program support costs such as staff training and supervision in our analysis—these were assumed to be common to both study arms and so omitted from the incremental analysis but can contribute a significant fraction of total intervention costs [37].

Alternative approaches should remain under consideration to reduce caregiver burden while promoting clinical recovery. Possible modifications could include adopting a visit frequency between the approaches tested in this trial (such as weekly follow-up until initial weight gain is achieved followed by monthly follow-up until discharge), basing the frequency of follow-up on an initial clinical risk assessment, or allowing caregivers to play a greater role in choosing the frequency of follow-up given full information of the various trade-offs. It is possible that these or other modifications could improve efficiency without jeopardizing health outcomes.

This study has several limitations. Firstly, while these results provide evidence against the monthly follow-up approach as a replacement for standard-of-care in this setting, they do not describe how these results might generalize to other settings. In particular, the health outcomes observed for nutritional recovery under both arms in this study were below international standards [38] and may be influenced by contextual factors including poor access to care or

suboptimal home use of RUTF (as discussed by Hitchings et al. [15]), which may vary across settings. It is unclear how the cost-effectiveness outcomes reported in this study would change in settings with a higher fraction of children achieving nutritional recovery. It is possible that different cost-effectiveness results would be obtained in settings where the costs of therapeutic food represent a smaller fraction of total costs or where the health risks associated with less frequent follow-up are lower. Moreover, factors beyond cost-effectiveness (feasibility, acceptability to caregivers) should be considered when selecting the intervention approach. These factors might favor monthly follow-up and could be influential in situations where the cost and health differences between monthly follow-up and standard-of-care are less stark. Secondly, our assessment of health outcomes only collected empirical data up to 3 months after the completion of treatment and modelled health outcomes for an additional 12 months. While it is unlikely that assessing outcomes over a longer time period would change study findings in a major way, doing so would capture any ongoing health effects or differences in healthcare utilization beyond the intervention period. Recent systematic reviews [4, 5] have demonstrated poorer long-term outcomes among children surviving SAM compared to those without a history of SAM, suggesting that more effective SAM treatment may ameliorate these long-term effects. Thirdly, it is possible that the present analysis did not fully capture all benefits of the reduced number outpatient consultations for SAM treatment achieved by the monthly follow-up approach. In theory, additional staff time due to fewer consultations per patient in the monthly follow-up approach could be spent treating additional SAM patients (increasing program coverage), treating more patients with other conditions, or improving quality of care. As is conventional for cost-effectiveness analysis, we assumed the reduced staff time required to provide monthly follow-up would produce cost savings for the SAM program (which are subtracted from the numerator of the cost-effectiveness ratio) rather than additional health benefits (which would be added to the denominator of the cost-effectiveness ratio). While the indirect health benefits of the monthly follow-up approach are unknown, if they could be quantified, it is possible they would produce more favorable cost-effectiveness results for the monthly follow-up approach than presented here. Fourthly, we did not compare the study arms to a 'no SAM program' counterfactual. This would have been unethical within the context of the study, yet the comparison itself is not implausible, as SAM treatment is not always possible in all settings. Studies that have assessed the cost-effectiveness of community-based SAM treatment (as compared to no SAM program) have found it to be highly cost-effective in a range of settings [29, 34–36]. In situations where a SAM program cannot be implemented, some version of the monthly follow-up approach may be practical and would very likely be cost-effective, given the high mortality associated with untreated SAM.

## Conclusion

The trial on which this economic analysis is based was conducted in a resource-constrained and rural area in northwestern Nigeria, and designed to test whether uncomplicated SAM could be effectively treated with a lower intensity of clinical follow-up. If this approach had been found to be effective and efficient, it would have allowed greater flexibility in the delivery of nutritional treatment in areas with high disease burden and limited healthcare access. While it was hypothesized that monthly follow-up could achieve equivalent health outcomes at lower cost, we found that this strategy led to both worse health outcomes and higher costs. Generalization of these findings should proceed with caution given the contextual factors that may have influenced health outcomes in this single study. Concerns related to the burden that weekly visits place on both caregivers and providers and could potentially be alleviated by monthly follow-up remain valid, but the findings from this study suggest that alternative

approaches may be needed to address this issue. Future modifications to the treatment protocols for uncomplicated SAM will need to be planned with a full appreciation of the health consequences that could result if the level of clinical care is insufficient and the need to make efficient use of high-cost intervention inputs.

## Supporting information

**S1 Checklist. CHEERS checklist.**
(PDF)

**S1 Table. Effect of monthly schedule of follow-up compared to standard weekly schedule of follow-up on primary and secondary effectiveness outcomes assessed at program discharge.**
(DOCX)

**S1 Protocol. Trial protocol.**
(PDF)

**S2 Protocol. Cost-effectiveness protocol.**
(PDF)

## Acknowledgments

We thank all the families and children who participated in this study; our field research teams; coordinators of the Médecins sans Frontières—Operational Center Amsterdam field mission in Nigeria; and members of the data and safety monitoring board (Saskia de Pee, André Briend, and Christopher Mambula).

## Author Contributions

**Conceptualization:** Nicolas A. Menzies, Fatou Berthé, Philip Aruna, Muhammed Ali Hamza, Siméon Nanama, Chizoba Steve-Edemba, Ibrahim Shehu, Rebecca F. Grais, Sheila Isanaka.

**Formal analysis:** Nicolas A. Menzies, Matt Hitchings.

**Funding acquisition:** Sheila Isanaka.

**Methodology:** Nicolas A. Menzies.

**Supervision:** Fatou Berthé, Philip Aruna, Muhammed Ali Hamza, Siméon Nanama, Chizoba Steve-Edemba, Ibrahim Shehu, Rebecca F. Grais, Sheila Isanaka.

**Writing – original draft:** Nicolas A. Menzies.

**Writing – review & editing:** Fatou Berthé, Matt Hitchings, Philip Aruna, Muhammed Ali Hamza, Siméon Nanama, Chizoba Steve-Edemba, Ibrahim Shehu, Rebecca F. Grais, Sheila Isanaka.

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
