## [Decision Letter · Decision Letter 0]

6 Sep 2022

PGPH-D-22-00813

Cost-effectiveness of monthly follow-up for the treatment of uncomplicated severe acute malnutrition: an economic evaluation alongside a randomized controlled trial.

Dear Dr. Menzies,

Thank you for submitting your manuscript to PLOS Global Public Health. After careful consideration, we feel that it has merit but does not fully meet PLOS Global Public Health’s publication criteria as it currently stands. Therefore, we invite you to submit a revised version of the manuscript that addresses the points raised during the review process.

EDITOR: Please insert comments here and delete this placeholder text when finished. Be sure to:

Thank you for submitting your paper to PGPH for publication. Two independent reviewers have assessed your manuscript and found it to have merit. However, they raised substantial methodological issues that will need your attention. The issues include:

-Clarity on the quantity of therapeutic food per participant, and the extended period of visits.

-They also felt there is a need for you to look at the definition of nutritional recovery.

-The stage at which participants were discharged.

-Justification of sample size used in the study.

-The assumptions made in the sample size estimation.

-There is also a need for clarity on the presentation of the results.

Please ensure that your decision is justified on PLOS Global Public Health’s publication criteria and not, for example, on novelty or perceived impact.

We look forward to receiving your revised manuscript.

Kind regards,

Dickson Abanimi Amugsi, PhD

Academic Editor

Journal Requirements:

1. 1. In the online submission form, you indicated that "The deidentified dataset supporting this research can be made available following a submitted request as per Epicentre and General Data Protection Regulation (EU) 2016/679 data sharing policy." All PLOS journals now require all data underlying the findings described in their manuscript to be freely available to other researchers, either 1. In a public repository, 2. Within the manuscript itself, or 3. Uploaded as supplementary information.

Additional Editor Comments (if provided):

Thank you for submitting your paper to PGPH for publication. Two independent reviewers have assessed your manuscript and found it to have merit. However, they raised substantial methodological issues that will need your attention. The issues include:

-Clarity on the quantity of therapeutic food per participant, and the extended period of visits.

-They also felt there is a need for you to look at the definition of nutritional recovery.

-The stage at which participants were discharged.

-Justification of sample size used in the study.

-The assumptions made in the sample size estimation.

-There is also a need for clarity on the presentation of the results.

Reviewers' comments:

Reviewer's Responses to Questions

**Comments to the Author**

1. Does this manuscript meet PLOS Global Public Health’s publication criteria? Is the manuscript technically sound, and do the data support the conclusions? The manuscript must describe methodologically and ethically rigorous research with conclusions that are appropriately drawn based on the data presented.

Reviewer #1: Partly

Reviewer #2: Yes

2. Has the statistical analysis been performed appropriately and rigorously?

Reviewer #1: Yes

Reviewer #2: Yes

3. Have the authors made all data underlying the findings in their manuscript fully available (please refer to the Data Availability Statement at the start of the manuscript PDF file)?

Reviewer #1: Yes

Reviewer #2: Yes

4. Is the manuscript presented in an intelligible fashion and written in standard English?

Reviewer #1: Yes

Reviewer #2: Yes

5. Review Comments to the Author

Reviewer #1: This paper presents an economic evaluation of monthly vs weekly follow-up treatment for malnutrition, based on the findings of a recent randomized controlled trial. Understanding the cost tradeoffs in malnutrition treatment and prevention is an important topic, and this paper provides valuable, and somewhat counter-intuitive evidence on an aspect of great importance in operating a malnutrition treatment program, namely that monthly follow-up produced worse outcomes at a higher cost than weekly followup.

There is one major methodological issue which I would like to better understand before I can determine the validity of this work, however. On line 158, the authors define their outcome of nutritional recovery as being free from complications and with MUAC of at least 125 mm for 2 consecutive visits. But it seems to me that since the treatment variable itself is the time between successive visits, this criteria might inadvertently lead to the treatment arm appearing less efficacious than the control arm, where there are successive visits closer together towards the end of the study. It appears possible, at least in theory, that this requirement could produce a bias the classification of recovery for the monthly treatment arm. As a thought experiment, consider if instead, caregivers had taken at-home measurements and reported MUAC at least 125 mm for two consecutive weeks to define recovery. If there are only three follow-up visits for the monthly follow-up arm, children must have a time-to-recovery of less than 8 weeks to be classified as recovered in this arm but could take up to 11 weeks to recover in the standard-of-care arm. Has this risk been addressed in the RCT in a way that could be explained in this paper?

Other Major Suggestions:

Line 362: I find it surprising that monthly approach required so many more sachets of therapeutic food. Your explanation about the extended period between visits is not totally satisfying to me. Is this because there is also a longer time-to-recovery in the monthly treatment arm, given the lower recovery rates, or simply because of wastage after recovery since the family has been supplied with a months worth of food? Is it perhaps not only that more food is distributed, but also that more food is required and/or consumed in the treatment arm of the trial? Is there more wastage or loss, e.g. as the family shares with relatives and neighbors when they have a month’s worth of sachets in their house. Any expansion on theorizing why this surprising finding has happened would be a welcome addition to the discussion.

Line 414: The point about the risk of generalizing these results to other settings (line 380), given the low recovery rates in both study arms seems important enough to include in the conclusion.

Line 374-377: I recommend that you include the costs of training and supervision if you can. Even if they are common to the treatment and control arm, it would be a missed opportunity if readers of your paper could see the scale of these essential parts of a successful program in context. It would be great to also include them in Table 1, for busy readers who might miss them in the text.

Minor issues:

Line 88: typo, should be “enrollment”

Line 91: Why does monthly follow-up include a visit at 10 weeks? Perhaps “monthly” is a confusing term for this less frequent visit schedule.

Line 91: Do these visits always end at 12 weeks? If there is a strict limit on the duration of the treatment, after which point children who are not recovered are still discharged from treatment, I would like this to be described in more detail.

Line 132: Can you describe in more detail how you determine the average visit duration?

Line 159: Please define precisely what it means for a child default.

Line 168: Please inform the reader of the time-horizon for the expected mortality after the trial here. (Is it 12 months?)

Line 565: typo, should be “not conducted” instead of “no conducted”.

Figure 1A, 1B, and 2A: Consider anchoring axes at zero to contextualize the difference between study arms relative to the absolute magnitude of each study arm --- this might tell a different story about the value of monthly follow-up in a setting where weekly follow-up is simply infeasible.

Figure 2B: I recommend excluding this figure, since the treatment is never cost-effective.

Reviewer #2: Cost-effectiveness of monthly follow-up for the treatment of uncomplicated severe acute malnutrition: an economic evaluation alongside a randomized controlled trial.

Overall this is an important paper and well worthy of publication in a high profile journal like PLoS Global Health. It is a well written report of a well done and robustly analysed study.

I have a few minor comments which I hope will help future readers make the most of the study:

- it’s not very clear (in title, abstract or final sentence of intro), whether this paper presents in-depth economic evaluation of a RCT that’s reported elsewhere (as suggested by reference 14 and by line 106) OR whether the cost-effectiveness was the major RCT outcome and this is the main paper reporting all outcomes. Please clarify. If the former, it’d be better to say this was an “economic evaluation OF a (separately reported) RCT”. It’d also be good in this case to briefly summarise the main RCT results in the methods.

The word “alongside” is what leads to the ambiguity – change that.

Intro:

- Consider giving some latest WHO/UNICEF numbers of children affected by SAM worldwide. This would help emphasise the importance of the study to readers not familiar with the field.

- Also consider citing the increasing evidence (e.g. two recent systematic reviews in BMJ Global Health) about the long term adverse effects of childhood SAM. This would also be worth including in the discussion as better treatment might have long term benefits: hence the costs of treatment might in fact have much more long lasting benefit and be even more cost-effective than appears in short-term analysis such as this.

Methods

- Lines 90-95. Did participants attend until 12 weeks irrespective of growth/progress or could they be discharged earlier if meeting specified anthro criteria (e.g. WLZ >-2 z-score on 2 consecutive visits – as is standard in many CMAM programmes worldwide. Please give details for both arms as again this could make a big difference to final costs.

- Line 100-107. Please summarise sample size rationale/justification. What outcomes was the trial powered to be able to observe? Did power calcs consider economic outcomes and if so what assumptions were made in the sample size calcs for this?

- Line 156 – assuming this is a economic analysis of a main trial reported elsewhere, re-phrase this section to make this clear. Consider also including a summary of the main trial results here.

- Line 158 – good to mention these discuarge criteria earlier as per previous comment. Was discharges only based on MUAC or was weight-for-length recovery also considered?

Results

- Even though this is an econ evaluation, a study flow chart would be helpful in first para of results – to get a sense of completeness of data and balance of drop-out/default between study arms

- Would be good if the main effectiveness results from the main paper are summed up previously. But if not, should do so here in the first para rather than much further down

- Line 211 – please also mention average in-programme stay (in weeks) in this section...this would help put the number of consults into context.

- Line 217 – rather than “non-significant” better to just cite the p-value

- Line 276 - % recovery seems very low in both arms – important to discuss this later in discussion. Was for example % default very high? A table summarising all standard CMAM effectiveness outcomes would be helpful (perhaps as online annex so readers don’t have to search for the main paper, ref 14)

Discussion

- Why was a monthly follow-up chosen? It’s quite a big jump from weekly to monthly – with hindsight (and for future research, esp in light of the observation re RUTF accounting for the greater costs) perhaps 2 or 3weekly follow-up would have been a better comparator against the “standard” weekly follow-up. Add to discussion/recommendation for future research

- Shorten conclusions section to be clearer what the study found. Do this by putting some of the text currently there into a “recommendations for future research” section in the prior discussion.

6. PLOS authors have the option to publish the peer review history of their article (what does this mean?). If published, this will include your full peer review and any attached files.

**Do you want your identity to be public for this peer review?** For information about this choice, including consent withdrawal, please see our Privacy Policy.

Reviewer #1: **Yes: **Abraham D. Flaxman

Reviewer #2: No

---

## [Decision Letter · Decision Letter 1]

8 Nov 2022

Cost-effectiveness of monthly follow-up for the treatment of uncomplicated severe acute malnutrition: an economic evaluation of a randomized controlled trial.

PGPH-D-22-00813R1

Dear Dr Menzies,

We are pleased to inform you that your manuscript 'Cost-effectiveness of monthly follow-up for the treatment of uncomplicated severe acute malnutrition: an economic evaluation of a randomized controlled trial.' has been provisionally accepted for publication in PLOS Global Public Health.

Best regards,

Dickson Abanimi Amugsi, PhD

Academic Editor

Thank you for adequately addressing the reviewers' comments. However, one of the reviewers raised a typo issue that I suggest you liaise with the editorial office to have it corrected, if you felt it is necessary to do so. See the issue below:

"I can’t resist making one additional minor suggestion, which is to swap the axes on Figure 3, so that DALYs are on the x-axis and costs are on the y-axis. In this axis-swapped version, the slope of a line has units that match the ICER (dollars per DALY)".

Reviewer Comments (if any, and for reference):

Reviewer's Responses to Questions

**Comments to the Author**

1. If the authors have adequately addressed your comments raised in a previous round of review and you feel that this manuscript is now acceptable for publication, you may indicate that here to bypass the “Comments to the Author” section, enter your conflict of interest statement in the “Confidential to Editor” section, and submit your "Accept" recommendation.

Reviewer #1: All comments have been addressed

Reviewer #2: All comments have been addressed

2. Does this manuscript meet PLOS Global Public Health’s publication criteria? Is the manuscript technically sound, and do the data support the conclusions? The manuscript must describe methodologically and ethically rigorous research with conclusions that are appropriately drawn based on the data presented.

Reviewer #1: Yes

Reviewer #2: Yes

3. Has the statistical analysis been performed appropriately and rigorously?

Reviewer #1: Yes

Reviewer #2: Yes

4. Have the authors made all data underlying the findings in their manuscript fully available (please refer to the Data Availability Statement at the start of the manuscript PDF file)?

Reviewer #1: Yes

Reviewer #2: Yes

5. Is the manuscript presented in an intelligible fashion and written in standard English?

Reviewer #1: Yes

Reviewer #2: Yes

6. Review Comments to the Author

Reviewer #1: This revision has addressed all of my questions and suggestions from the previous draft, and I recommend swift publication of this work.

I can’t resist making one additional minor suggestion, which is to swap the axes on Figure 3, so that DALYs are on the x-axis and costs are on the y-axis. In this axis-swapped version, the slope of a line has units that match the ICER (dollars per DALY).

Reviewer #2: Thank you for making these edits and modification. The paper is even better as a result. I really enjoyed reading and learning from your work and I'm sure many others will do likewise. This will be an excellent contribution to the literature. Good luck with future work on this important topic of child malnutrition.

7. PLOS authors have the option to publish the peer review history of their article (what does this mean?). If published, this will include your full peer review and any attached files.

**Do you want your identity to be public for this peer review?** For information about this choice, including consent withdrawal, please see our Privacy Policy.

Reviewer #1: **Yes: **Abraham D. Flaxman

Reviewer #2: No
